# Surface Marker Expression in Small and Medium/Large Mesenchymal Stromal Cell-Derived Extracellular Vesicles in Naive or Apoptotic Condition Using Orthogonal Techniques

**DOI:** 10.3390/cells10112948

**Published:** 2021-10-29

**Authors:** Renata Skovronova, Cristina Grange, Veronica Dimuccio, Maria Chiara Deregibus, Giovanni Camussi, Benedetta Bussolati

**Affiliations:** 1Department of Molecular Biotechnology and Health Sciences, University of Turin, 10126 Turin, Italy; renata.skovronova@unito.it (R.S.); veronica.dimuccio@unito.it (V.D.); 2Department of Medical Sciences, University of Turin, 10126 Turin, Italy; cristina.grange@unito.it (C.G.); giovanni.camussi@unito.it (G.C.); 32i3T Business Incubator and Technology Transfer, University of Turin, 10126 Turin, Italy; mariachiara.deregibus@unito.it

**Keywords:** tetraspanins, 10k MSC EVs, 100k MSC EVs, Nanosight, MACSPlex, ExoView, super-resolution microscopy

## Abstract

Extracellular vesicles released by mesenchymal stromal cells (MSC-EVs) are a promising resource for regenerative medicine. Small MSC-EVs represent the active EV fraction. A bulk analysis was applied to characterise MSC-EVs’ identity and purity, with the assessment of single EV morphology, size and integrity using electron microscopy. We applied different methods to quantitatively analyse the size and surface marker expression in medium/large and small fractions, namely 10k and 100k fractions, of MSC-EVs obtained using sequential ultracentrifugation. Bone marrow, adipose tissue and umbilical cord MSC-EVs were compared in naive and apoptotic conditions. As detected by electron microscopy, the 100k EV size < 100 nm was confirmed by super-resolution microscopy and ExoView. Single-vesicle imaging using super-resolution microscopy revealed heterogeneous patterns of tetraspanins. ExoView allowed a comparative screening of single MSC-EV tetraspanin and mesenchymal markers. A semiquantitative bead-based cytofluorimetric analysis showed the segregation of immunological and pro-coagulative markers on the 10k MSC-EVs. Apoptotic MSC-EVs were released in higher numbers, without significant differences in the naive fractions in surface marker expression. These results show a consistent profile of MSC-EV fractions among the different sources and a safer profile of the 100k MSC-EV population for clinical application. Our study identified suitable applications for EV analytical techniques.

## 1. Introduction

Mesenchymal stromal cells (MSCs) are nowadays the most commonly used cell source for regenerative medicine due to their immunomodulatory, pro-regenerative and anti-inflammatory properties [1,2,3]. MSCs can originate from different tissues, including bone marrow (BM), adipose tissue (AT) and the umbilical cord (UC), which the ones most commonly used. As the field of extracellular vesicles (EVs) rises, so does the interest in the isolation and therapeutic application of MSC bioproducts. Indeed, MSC EVs may overlap many of the described effects of the originating cells [4], playing an essential role in cell-to-cell communication, involving stem cells and targeting injured cells [5,6].

EVs are released from cells as a heterogeneous population that can be further classified into three fractions: small EVs, large EVs and apoptotic bodies (ApoBDs) based on their size and composition, with ranges defined either as <100 nm or <200 nm for small EVs, and >200 nm for medium/large EVs as the MISEV community classified [7]. Moreover, the classification in different EV fractions underlines distinct molecular and functional properties [8]. In addition, the EVs can also be classified into exosomes and microvesicles depending on their biogenesis from multivesicular bodies or cell surface budding, respectively [7]. However, small and medium/large EVs share many structural components. A specific surface marker expression of EV fractions lacks a strict boundary between them due to the absence of a strict boundary [9]. In particular, the classical tetraspanins, CD9, CD63 and CD81, are commonly present in different EV subpopulations [7]. However, a higher expression level on small MSC EVs was reported, in line with their possible origin in the cellular endosomal pathway [7]. At variance, the large EV surface may resemble the parental cell origin more closely, as it can be likely associated with direct budding from the plasma membrane [10]. Moreover, medium/large EVs are generally enriched with phosphatidylserine [10] and CD40 [11]. The therapeutic effect of MSC EVs was first ascribed to the entire heterogeneous EV population released by the cells under culture conditions. Subsequent studies, however, tried to identify the potentially most relevant subpopulation by fractionating MSC EVs in medium/large EVs (100–1000 nm) using a 10,000× *g* ultracentrifugation (10k fraction) and in small MSC EVs (<100 nm) using a subsequent 100,000× *g* ultracentrifugation (100k fraction). In vitro and in vivo pre-clinical experiments clearly showed that the 100k fraction was the main fraction responsible for functional and morphological tissue regeneration [12,13,14]. Indeed, the 10k and 100k fractions appeared biochemically and functionally distinct [12,15]. The small MSC EVs nowadays consider the proactive fraction retaining the therapeutic activity [16].

The characterisation of the small therapeutic MSC EVs required to fulfil standard EV analyses, including evaluation of morphology, size and expression of vesicular and non-vesicular markers in accordance with the minimal information for studies of extracellular vesicles 2018 (MISEV) [7], coupled with the presence of typical MSC surface antigens and lack of non-MSCs markers, reflects the identity criteria defined for the originating cells by the International Society for Cell & Gene Therapy minimal criteria [16,17]. Indeed, it is of interest and of potential relevance for clinical application to determine and quantify the expression of identity markers such as tetraspanins and mesenchymal markers, as well as of other immunological and pro-coagulative surface markers within the small MSC EV population as compared with medium/large EVs in MSCs of different origin.

Furthermore, recent studies identify that the MSC-mediated immunomodulatory effects in vivo are due to apoptosis, suggesting a therapeutic role for apoptotic EVs [18]. However, knowledge of the differences between apoptotic and naive MSC EVs is still limited. In the present work, we aimed to determine the surface marker expression of small MSC EVs isolated by sequential ultracentrifugation at 100,000× *g* (after removal of the 10,000× *g* centrifugation), defined here as 100k MSC EVs, as compared to medium/large MSC EVs isolated at 10,000× *g* and defined as 10k MSC EVs. In particular, we aimed to characterise and compare the profile of EVs from three different MSC sources of clinical interest, applying the same experimental conditions for MSC culture, EV isolation and analysis. For this comparison, we used different techniques following the standards and requirements of the ISEV community, including innovative single-EV analysis techniques such as ExoView chip and super-resolution microscopy, as well as bead-based cytofluorimetric analysis. Standard culture and apoptotic conditions were applied.

## 2. Materials and Methods

### 2.1. Cell Culture

The MSCs were obtained in collaboration with the RenalToolBox ITN (Grant Agreement 813839). BM-MSCs obtained by the group of Prof. Timothy O’Brien (National University of Galway, Galway, Ireland) were purchased from Lonza (Basel, Switzerland), AT-MSCs from lipoaspirate adipose tissue harvested processed by the group of Prof. Karen Bieback (University of Heidelberg, Heidelberg, Germany) after informed consent. The Mannheim Ethics Commission II approved the study (vote 2011-215N-MA). UC-MSCs were obtained from the group of Dr Jon Smythe (NHS Blood and Transplant, Liverpool, UK) from three different healthy donors after informed consent, as per the approved protocol of the NHS Blood and Transplant Unit. MSCs were cultured and expanded under standardised protocol among the groups. In particular, all MSCs were cultured using AlphaMEM with UltraGlutamine (BE02-002F, Lonza, Basal, Switzerland) and 10% Foetal Bovine Serum (10270-106, Gibco, MA, USA) in the incubator at 37 °C with 5% CO_2_ and controlled humidity. MSCs were checked for the expression of mesenchymal markers by cytofluorimetric analysis (data not shown). EVs were collected from MSCs at 4-6th passage.

### 2.2. EV Isolation

When the cells reached 80% confluency, they were starved overnight (16 h) in RPMI medium (Figure 1). The supernatant was collected and centrifuged for 10 min at 300× *g* to remove cell debris on the second day. In experiments using apoptotic MSCs, the supernatant was transferred into new tubes and centrifuged 3000× *g* for 20 min to collect apoptotic bodies. The supernatant was then ultracentrifuged for 1 h at 10,000× *g*, 4 °C, using the Beckman Coulter Optima L-100K Ultracentrifuge (Beckman Coulter, CA, USA) with the rotor type 70Ti. At this speed, the subpopulation of 10k EVs was collected. The supernatant was further ultracentrifuged for 1 h at 100,000× *g*, 4 °C to obtain the 100k EV subpopulation. The EV pellet was resuspended in RPMI supplemented with 0.1% DMSO. The EV suspension was then stored at −80 °C until further use.

Apoptotic vesicles were isolated from MSCs undergoing apoptosis for 24 h using 500 ng/mL of an Anti-Fas Ab (3510771, Merck, NY, USA) diluted in RPMI medium. Isolation of the apoptotic EVs followed the same ultracentrifugation protocol of the naive MSC EVs. The ApoBDs were re-suspended in RPMI supplemented with 1% DMSO.

### 2.3. Nanoparticle Tracking Analysis

After the isolation, the concentration of all the samples was measured by Nanosight NS300 (Malvern Instruments Ltd., Malvern, UK) equipped with a 488 nm laser module that utilises Brownian motion and refraction index. The particle size scatters 10 nm to 1000 nm, although the optimised size range is 70–300 nm. It uses the scattered light to detect a particle and tracks its motion as a function of time. The particles’ scattered light was recorded with a light-sensitive camera under a 90° angle to the irradiation plane. This angle allows the Brownian motion of the EVs. Samples were diluted 1:100 in physiologic solution. For each sample, 3 videos of 60 s at camera level 15 and threshold 5 were captured using a syringe pump 30. All the samples were characterised with NTA 3.2 Analytical software. The NTA settings were kept constant between samples.

### 2.4. Transmission Electron Microscopy

The transmission electron microscopy (TEM) was performed on EVs placed on 200-mesh nickel formvar carbon-coated grids (Electron Microscopy Science) for 20 min to promote adhesion. The grids were then incubated with 2.5% glutaraldehyde plus 2% sucrose. EVs were negatively stained with NanoVan (Nanoprobes, Yaphank, NY, USA) and observed using a Jeol JEM 1400 Flash electron microscope (Jeol, Tokyo, Japan).

### 2.5. Cytofluorimetric Analysis

MACSPlex Exosome Kit (Miltenyi Biotec, Bergisch Gladbach, Germany) containing fluorescently labeled (FITC-PE) capture beads coupled to 37 exosomal surface epitopes and 2 isotope controls was used, following the manufacturer’s instructions (in detail: CD3, CD4, CD19, CD8, HLA-DR, CD56, CD105, CD2, CD1c, CD25, CD49e, ROR1, CD209, CD9, SSEA-4, HLA-ABC, CD63, CD40, CD62P, CD11c, CD81, MCSP, CD146, CD41b, CD42a, CD24, CD86, CD44, CD326, CD133-1, CD29, CD69, CD142, CD45, CD31, REA control, CD20, CD14, mIgG1 control). Briefly, 15 µL of beads were added to 120 µL of buffer or sample, including a total of 1 × 10^9^ EVs, and the complex was then incubated on a rotor overnight at 4 °C. After the incubation and washing steps, a cocktail of APC fluorescent antibodies against tetraspanins (CD9, CD63 and CD81) was added (allowing the detection of beads-bound EVs) and set on the rotor for 1 h at room temperature. After washing, samples were detected using BD FACSCelesta™ Flow Cytometer (BD Bioscience, NJ, USA). Median background values of buffer control were subtracted, and samples were normalised to the median fluorescence intensity of tetraspanins.

### 2.6. ExoView Chip-Based Analysis

NanoView Biosciences (Boston, MA, USA) customised silicone chips coated with tetraspanins, CD44 and CD105 were incubated overnight with 1 × 10^8^ MSC EVs suspension diluted in a final volume of 35 µL of incubation buffer A at room temperature. After the incubation, chips were washed 3 times for 3 min on an orbital plate shaker with wash solution B. The chips were scanned with the ExoView™ R100 reader (NanoView Biosciences) by the ExoScanner software (3.0, NanoView Biosciences, Boston, MA, USA). The particle size scatters 50 nm to 200 nm. The data were analysed using ExoViewer software (3.0, NanoView Biosciences, Boston, MA, USA). The number of captured EVs for each surface epitope were compared between the samples.

### 2.7. Super-Resolution Microscopy

Super-resolution microscopy pictures of EVs were obtained using a temperature-controlled Nanoimager S Mark II microscope from ONI (Oxford Nanoimaging, Oxford, UK) equipped with a 100x, 1.4NA oil immersion objective, an XYZ closed-loop piezo 736 stage, and 405 nm/150 mW, 473 nm/1 W, 560 nm/1 W, 640 nm/1 W lasers, as well as dual/triple emission channels split at 640 / and 555 nm. The samples were prepared using 10 μL of 0.01% Poly-L-Lysine (Sigma-Aldrich, St. Louis, MO, USA) placed on high-precision coverslips cleaned in sonication bath 2 times in dH_2_O and once in KOH, in silicon gasket (Sigma-Aldrich, St. Louis, MO, USA). The coated coverslips were placed at 37 °C in a humidifying chamber for 2 h. Excess of Poly-L-Lysine was removed. Then, 1 μL of EVs (1 × 10^10^) resuspended in 9 μL of blocking solution (PBS-5% Bovine Serum Albumin) were pipetted into a previously coated well to attach overnight at +4 °C. The next day, the sample was removed, and 10 μL of blocking solution was added into the wells for 30 min. Antibodies were directly conjugated as follows: 2.5 μg of purified mouse anti-CD9 was conjugated with Atto 488 dye (ONI, Oxford, UK), and anti-CD63, CD40 and Annexin V antibodies (Santa Cruz, CA, USA: SC-5275, SC-13128, SC-8300) were conjugated with Alexa Fluor 555 dye. Anti-CD81, Annexin A1 and Anti-Phosphatidylserine antibodies (Santa Cruz, CA, USA; SC-31234, SC-12740. Merck, NY, USA; 05-719) were conjugated with Alexa Fluor 647 dye using the Apex Antibody Labelling Kit (Invitrogen, Carlsbad, CA, USA) according to the manufacturer’s protocol. Samples were incubated with 1 μL of each antibody, added into blocking buffer at a final dilution 1:10, under light protection, overnight at +4 °C. The day after, samples were washed twice with PBS, and 10 μL of the mixed ONI B-Cubed Imaging Buffer (Alfatest, Rom, Italy) was added for amplifying the EV imaging. Two-channel (647 and 555) dSTORM data (5000 frames per channel) or three channels (2000 frames per channel) (647, 555 and 488) were acquired sequentially at 30 Hertz in total reflection fluorescence (TIRF) mode. Before each imaging session, beads slide calibration was performed to align fluorescent channels, achieving a channel mapping precision smaller than 12 nm. Single-molecule data was filtered using NimOS (Version 1.18.3, ONI, Oxford, UK) based on the point spread function shape, photon count and localisation precision to minimise background noise and remove low-precision and non-specific co-localisation. All pictures were analysed using algorithms developed by ONI via their CODI website platform (https://alto.codi.bio/, 3 October 2021). The filtering and drift correction were used as in NimOS software. The BDScan clustering tool was applied to merged channels, and co-localised EVs were also counted in separate channels.

### 2.8. Statistical Analysis

Data are shown as mean  ±  SD. At least three independent replicates were performed for each experiment. Statistical analysis was carried out on Graph Pad Prism version 8.04 (GraphPad Software, Inc., San Diego, CA, USA) by using the two-way ANOVA followed by Turkey’s multiple comparisons test, where appropriate. A *p* value < 0.05 was considered significant.

## 3. Results

### 3.1. Isolation of 100k and 10k MSC EVs and Size Analysis

Two MSC EV fractions were isolated using sequential centrifugations, as detailed in Methods; in particular, medium/large MSC EVs were isolated by a 10,000× *g* ultracentrifugation (10k fraction), followed by the small MSC EV isolation from the remaining supernatant by a 100,000× *g* ultracentrifugation (100k fraction) (Figure 1). MSCs were obtained from bone marrow, adipose tissue and the umbilical cord from three different donors for each cell source. To allow comparison among MSC EVs of different origin, we cultured MSCs in standardised superimposable conditions.

The 100k and 10k MSC EV fractions were first analysed using Nanoparticle tracking analysis (Figure 2A), which confirmed that 100k MSC EVs was a homogenous population. At variance, 10k EVs showed a multi-peak profile, indicating the presence of fractions with a highly variable size (Figure 2A). Transmission electron microscopy analysis showed the 100k EV morphology, as spherical, membrane-encapsulated particles with a characteristic cup-shaped aspect (Figure 2B). In contrast, the 10k EVs represented a heterogeneous population of EVs, differing greatly in size, shape and electron-density (Figure 2B). Quantitative size analysis using Nanoparticle tracking analysis showed that EV size (mode size) was superimposable between the different MSC EV sources (Table 1).

In addition, no differences were detected in the size of the 100k EVs in respect to the 10k EV fractions using this technique. In contrast, by electron microscopy, the majority of 100k EVs were smaller than 100 nm, whereas the majority of 10k EVs were in a size range of 100–300 nm (Table 2).

Using super-resolution microscopy based on tetraspanin staining (CD63 and CD81) on intact unfixed MSC EVs, we also confirmed the size of tetraspanin-expressing EV fractions (Figure 2C), being 100k EVs quantified, as around 90 nm median size for all MSC sources using an automatic size analysis software, whereas 10k EVs have a median size of around 130 nm (see Table 2). Single molecule analysis of CD63 expression also indicated its differential distribution on the EV surface, as 10k EVs showed a discrete surface expression, whereas 100k EVs showed a more condensed tetraspanin localisation, possibly due to the small size (Figure 2C). The patchy distribution of tetraspanins was more evident on larger EVs within the 10k fraction (>500 nm) (Appendix A).

#### Variable Tetraspanin Expression on 100k and 10k Single MSC EVs by Super-Resolution Microscopy

Transmembrane tetraspanin proteins CD63, CD9 and CD81 are a major class of EV-expressed molecules previously reported to be enriched in 100k in respect to 10k EV fraction [19,20]. We first took advantage of super-resolution microscopy to assess tetraspanin co-expression at a single EV level on 100k and 10k MSC EVs (Figure 3). Advanced three-colour staining was performed using the anti-tetraspanin Abs dyed in red (CD81), green (CD63) and blue (CD9) using dSTORM single-molecule analysis with super-resolution microscopy. Tetraspanin single-molecule surface analysis highlighted an uneven tetraspanins distribution on the EV surface. Moreover, we observed a heterogeneous tetraspanin distribution of EVs variably positive for single, double or triple tetraspanins (Figure 3). We also took advantage of an automated software analysis for the quantification of tetraspanin co-expression on single EVs (Figure 3C,D).

In particular, the triple tetraspanin expression only represented a fraction of the entire EV population, as the other EVs were variably positive for the different markers (Figure 4). The 100k MSC EV fraction, in general, did not show increased tetraspanin expression in respect to the 10k fraction. CD63 was the most expressed marker in the single positive EV population in AT- and UC-MSC EVs, but not in BM-MSC EVs (Figure 4). The 10k fraction of UC-MSC EVs showed the largest population of EVs co-expressing CD81, CD63 and CD9 (Figure 4B). These results show a variable co-expression of the tetraspanins on MSC EV sources, without significant differences in the 10k and 100k fractions.

### 3.2. Isolation of 100k and 10k MSC EVs from Apoptotic Cells

We also generated 100k and 10k EVs from MSCs undergoing apoptosis using an anti-Fas Ab, as described [21,22], for further comparison with the naive fractions. Fas triggered MSC apoptosis induction was confirmed by Annexin V staining (Appendix A). Moreover, apoptosis induction was also assessed by the generation of large apoptotic bodies (size range 1–5 µm), showing positivity for the apoptotic marker Annexin V by flow cytometry and by super-resolution microscopy (Appendix A). Moreover, apoptotic bodies also showed positivity for Phosphatidylserine (Appendix A), as previously described [7].

The size and number of apoptotic MSC EVs were analysed by nanoparticle tracking analysis and by electron microscopy, showing a similar size of the naive MSC EVs for both the 100k and 10k fractions (Table 1 and Figure 5). However, apoptotic 100k EVs showed a less homogeneous profile than the naive ones in the nanoparticle tracking analysis (Figure 5A). Interestingly, the average concentration of both 100k and 10k EVs from apoptotic cells was significantly higher than that obtained from naive cells using the same originating cell number (Table 1). Superimposable results were obtained for the three MSC sources. Using super-resolution microscopy, we detected the expression of Phosphatidylserine and Annexin V on apoptotic 100k and 10k and not on naive MSC EVs, confirming that these markers are able to specifically characterise the apoptotic EVs (Figure 5C,D) [23].

#### 3.2.1. Quantitative Tetraspanin Evaluation of Naive and Apoptotic 100k and 10k MSC EVs

We therefore used cytofluorimetric analysis and ExoView to gain quantitative results for the comparison of tetraspanin level expression in MSC EV fractions from all different sources, in naive and apoptotic MSC EVs.

Semiquantitative analysis of tetraspanin levels in all MSC EV subsets was performed using the MACSPlex exosome kit, a bead-based cytofluorimetric analysis. This technique did not detect differences among MSC sources, MSC EV fractions and naive or apoptotic conditions (Figure 6).

ExoView chip-based analysis was then used to obtain an evaluation of the number of the particles captured on a specific chip coated with tetraspanins and negative mouse IgG control (MIgG). The number of EVs loaded onto each chip was normalised based on their concentration, as evaluated with the nanoparticle tracking analysis. The results showed that the apoptotic EV fractions showed a higher number of tetraspanin-captured EVs than the naive ones, and that the 10k fractions showed a higher number of tetraspanin-captured EVs than the 100k fractions for all MSC sources used (Figure 7). Among the different sources the tetraspanin levels were significantly different, and UC-MSC EVs had the highest expression of most markers (Figure 7).

Comparing different techniques, CD9 expression by MACSPlex appeared lower than other tetraspanins for all MSC sources and fractions, at the variance of the results obtained using ExoView or super-resolution microscopy. These data highlight that ExoView provides a better quantitative analysis regarding the bead-based cytofluorimetric assay, performing a semiquantitative analysis. Moreover, different antibody affinities could be present.

#### 3.2.2. Mesenchymal, and Immunological Marker Expression on Naive and Apoptotic 100k and 10k MSC EVs

Mesenchymal markers are usually assessed to characterise MSC EVs [24]. The MACSPlex exosome kit allowed the evaluation of CD56, CD44, CD29, CD49e, CD146 and CD105 mesenchymal marker expression on all EV fractions, from the three MSC sources in naive and apoptotic conditions (Figure 8).

Mesenchymal markers were expressed from all MSC sources and MSC EV fractions in the naive and apoptotic EV conditions, but higher levels were generally observed in the 10k fraction with respect to the 100k fraction. We confirmed the expression of the mesenchymal CD105 and CD44 markers on the 100k naive and apoptotic MSC EVs (Figure 9A,B). CD44 and CD105 had a wider size range by size distribution of captured EVs, which is especially evident in the 100k apoptotic MSC EVs (Figure 9C,D).

Moreover, as demonstrated by the MACSPlex exosome kit, all EVs were negative for CD14, CD19, CD31 and CD45, as were the originating cells (data not shown). Interestingly, the 100k fractions were constantly negative for immunological markers, selectively expressed by the 10k EVs (Figure 10A–D). In particular, the fraction enriched for 10k EVs both in naive and apoptotic conditions was selectively expressing HLA-class I and CD40 co-stimulatory molecule. Moreover, tissue factor (TF), known to be involved in platelet activation, was expressed by 10k EVs of AT- and UC-MSC EVs, and not from those of BM-MSC EVs (Figure 10B,D).

## 4. Discussion

Small MSC EVs appear to be the most promising EV type for therapeutic application, and the information on the surface marker expression characterising the different MSC sources and fractions is of importance. This study presents a quantitative analysis of the surface expression profile of tetraspanins at a single EV level, showing variable tetraspanin coexpression in all EV fractions and sources by super-resolution microscopy. Moreover, using bead-based cytofluorimetric analysis and chip-based arrays, tetraspanins, as well as other clinically relevant markers (mesenchymal, immunological and pro-coagulative markers), were compared in MSC EVs from three sources in naive or apoptotic condition. The results highlight a similar characterisation profile of MSC EVs from the different MSC sources, with variable but consistent tetraspanin expression. Moreover, we observed an increased expression of mesenchymal surface markers and the restricted expression of HLA-class I, the co-stimulatory molecule CD40 and tissue factor by 10k MSC EVs, with respect to 100k MSC EVs. Finally, apoptotic conditions only modified the number, not the characterisation profile, of MSC EVs. Further functional analysis will be required to fully compare the properties of the different EV fractions and sources.

Thanks to an extensive effort of the EV community, the minimal criteria for EV characterisation have been set in open-access publication MISEV 2018 [7]. The analysis of the EV preparation includes a bulk analysis of protein expression demonstrating the EV identity and purity together with qualitative and quantitative analysis using a particle counter and electron microscopy [25]. The development of new nanotechnological instruments in recent years may allow the assessment of EV identity at a single EV level using affinity-based chips, super-resolution microscopy and high-resolution flow cytometry.

Here, we compared different orthogonal methods to provide a single EV analysis of 100k and 10k MSC EVs, highlighting their potential contribution and utility for 100k MSC EVs characterisation. The analysis of the EV size appeared discordant between the commonly used nanoparticle tracking analysis and the other methods. In fact, nanoparticle tracking analysis clearly showed a differential profile of 100k and 10k EVs, but the mean size of the 100k fraction was higher than that detected with other quantitative methods, as previously described [12]. This could be due to phenomena of EV aggregation, or to the influence of both temperature and Brownian motion incorporated in the nanoparticle tracking method of EV characterisation. In addition, particle size using this method can result in non-consistent data; similar analyses were reported with a difference of 15–50% in size [26].

Electron microscopy analysis clearly showed that the 100k MSC EVs and 10k MSC EVs had distinct size. Interestingly, this observation was in line with that obtained by the specific analysis of tetraspanin expressing EVs, acquired with super-resolution microscopy on more than 10,000 analysed fresh, unfixed EVs. In addition, quantitative single-vesicle imaging by super-resolution microscopy revealed a heterogeneous pattern of tetraspanin expressions (single, double and triple) in variable proportions in 100k and 10k MSC EVs. Recently, a single-vesicle imaging and co-localisation analysis of tetraspanins were investigated in EVs derived from HEK397, breast cancer and melanoma cell line, showing distinct fractions of single, double or triple co-expressing EVs, depending on the analysed EV type [27]. This is consistent with the observation that CD9- and CD81-positive EVs did not correlate with distinct EV populations using a size-based EV separation technique [28]. Another relevant feature was the patchy distribution of tetraspanins on the EV surface, usually characteristic of 10k EVs. Indeed, tetraspanins are known to homodimerize and form large complexes [29]. This may suggest the ability of tetraspanins on EV surface to move within the lipidic membrane, as described for the cell membrane, with capping after antibody binding [30].

Different MSC sources can be identified for clinical application of deriving EVs, the BM-MSC EVs being the first and most commonly used in clinical trials. However, EVs from adipose tissue and the umbilical cord might display advantages for easier cell isolation from adipose tissue, reduced impact of donor diseases or enhanced potency for umbilical cord [31]. We therefore compared 100k and 10k EVs from MSCs cultured in completely identical culture conditions (media and serum, expansion number and passages) for the expression of clinically relevant markers using a standard semiquantitative cytofluorimetric assay commonly used to assess MSC EV profile [32]. Interestingly, mesenchymal markers were present in both 100k MSC EVs and 10k MSC EVs, but with higher expression in the apoptotic EV fractions. At variance, HLA class I, co-stimulatory molecule and tissue factor expression were selectively expressed on 10k MSC EVs. These data suggest that the 100k MSC EV fractions are safe for clinical application, avoiding the possible development of anti-HLA antibodies and rejection. Moreover, 10k EVs from AT- or UC-MSCs also showed the expression of tissue factor. This is more in line with the increased coagulative capacity of 10k than that of the 100k MSC EV fractions previously reported [33]. Moreover, 10k BM-MSC EVs appeared to display the lowest expression of tissue factor, in analogy with reports showing higher thrombogenic activity of UC- or AT-MSCs in respect to BM-MSC EVs [34,35].

Finally, in the present study, we also compared naive MSC EVs and apoptotic MSC EVs from the different MSC sources. Indeed, apoptotic EVs are considered to display peculiar functions and are now considered an additional but distinct MSC product with potent immunoregulatory ability [18]. The EVs were released by apoptotic cells after Fas receptor triggering [21,22], and apoptosis was confirmed by the detection of apoptotic bodies and by expression of Phosphatidylserin [10] and Annexin V by apoptotic EVs [36]. The most relevant feature observed was the increase in number of both 100k and 10k MSC EVs released from all MSC types under apoptotic conditions. Moreover, apoptotic EVs expressed higher tetraspanin levels and mesenchymal markers with respect to the normal counterpart, as evaluated by ExoView and MACSPlex analysis, respectively.

## 5. Conclusions

In conclusion, our results show that the characterisation profile of MSC EV fractions is consistent among different MSC sources, with an increased number of EVs released under apoptotic condition. Moreover, the 100k MSC EV population displays a safer profile than the 10k MSC EV population for immunological and pro-coagulative marker expression. Finally, our study identified advantages of the different EV analytical techniques for specific applications. In particular, super-resolution microscopy was useful to characterise a large number of EVs at a single EV level, whereas ExoView analysis allowed an easy quantitative comparison of marker expression among fractions of different origins and conditions. In addition, bead-based cytofluorimetric analysis appeared to be of utility for its large variety of markers of clinical applicability, although it can only provide semiquantitative results. These results suggest that quantitative EV analysis methods are useful and reliable enough to be applied for the characterisation of MSC EV fractions.

## Figures and Tables

**Figure 1 cells-10-02948-f001:**
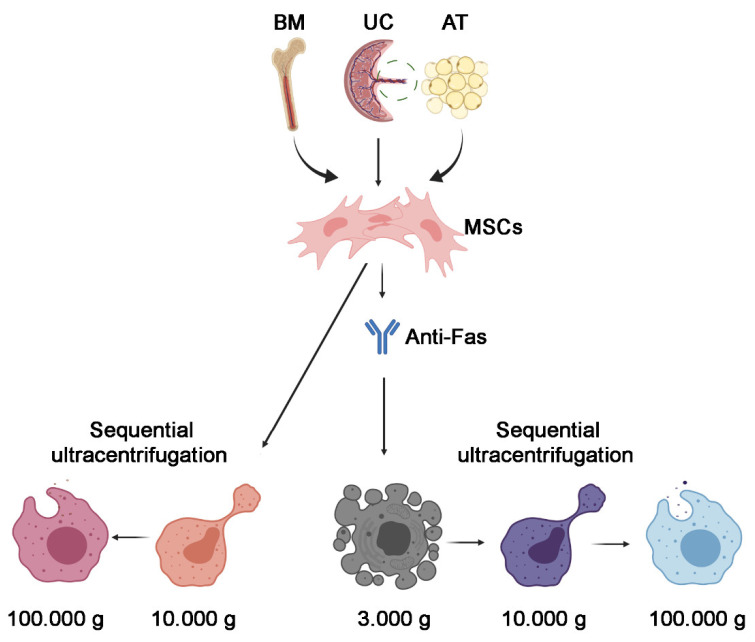
Scheme of different EV fractions used in this study. Naive and apoptotic MSC EVs, induced with anti-Fas antibody, obtained from bone marrow (BM), the umbilical cord (UC) or adipose tissue (AT) were isolated using subsequent differential ultracentrifugation. Figure was created using BioRender licence number BO233A7CUA.

**Figure 2 cells-10-02948-f002:**
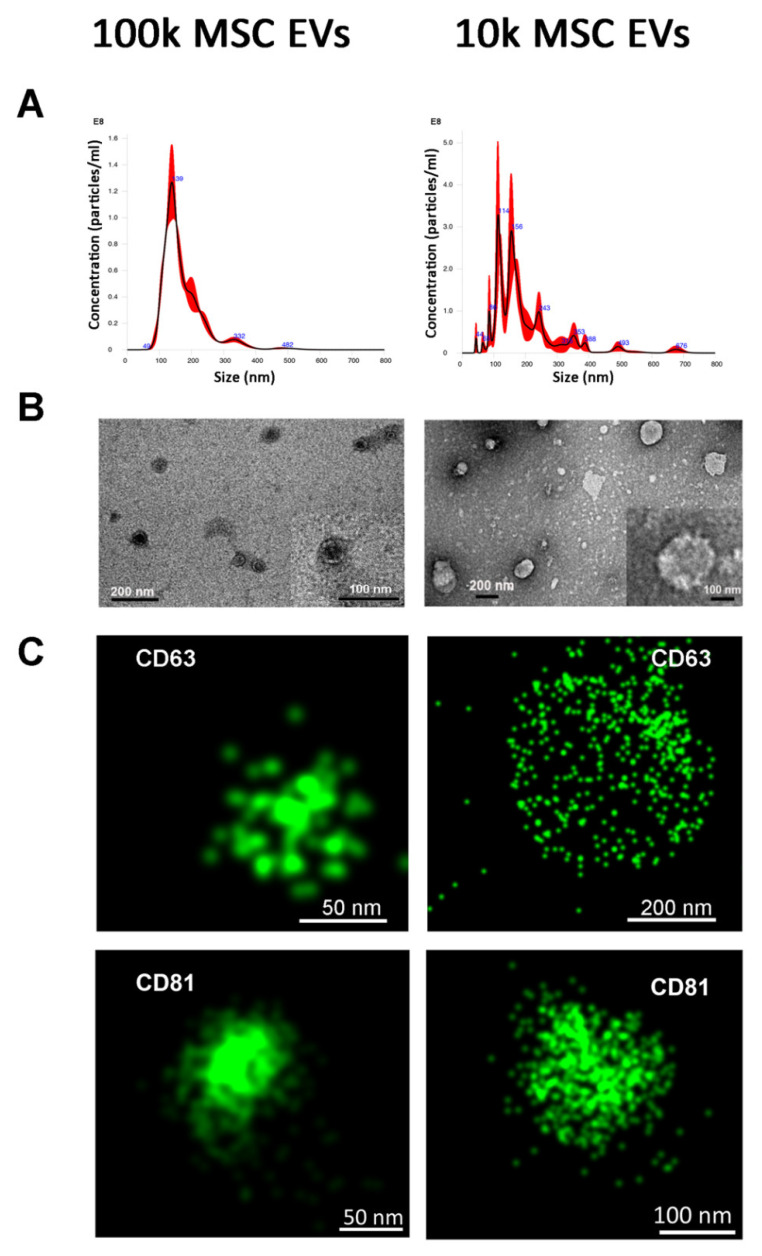
Characterisation of 100k and 10k naive MSC EVs. (**A**) Representative graphs of nanoparticle tracking analysis of 100k MSC EVs (left panel) and 10k MSC EVs (right panel). (**B**) Representative images of transmission electron microscopy of 100k MSC EVs (left panel) and 10k MSC EVs. The corresponding scale bare is below each EV image. (**C**) Representative super-resolution microscopy images of 100k MSC EVs (left panel) and 10k MSC EVs (right panel) stained with CD63 or CD81 tetraspanins.

**Figure 3 cells-10-02948-f003:**
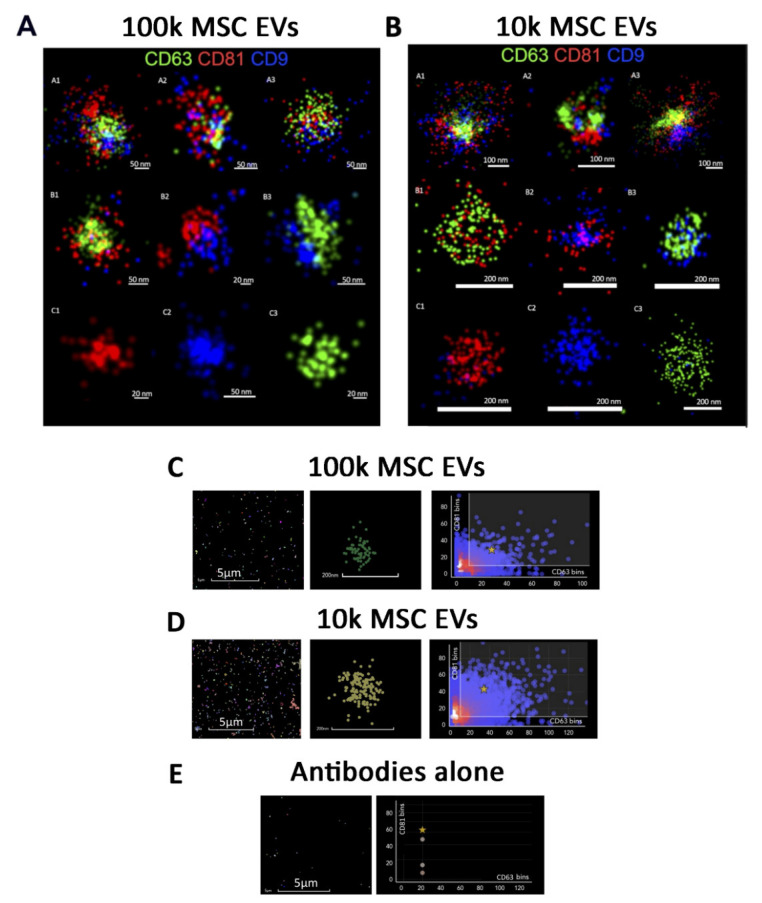
Super-resolution microscopy images. Representative super-resolution microscopy images of 100k EVs (**A**) and 10k EVs (**B**) stained with tetraspanins: CD81 red, CD63 green, and CD9 blue. Both panels (**A**,**B**) show triple positive, double positive and single positive MSC EVs. The corresponding scale bare is below each EV image. (**C**) Representative clustering strategy of MSC 100k EVs with large field of view (left panel), a selected cluster (central panel) and graph (right panel) showing CD81/CD63 cluster distribution. (**D**) Representative clustering strategy of MSC 10k EVs with large field of view (left panel), a selected cluster (central panel) and graph (right panel) showing the proportion of CD81 and CD63 antibodies/cluster. (**E**) Representative clustering strategy of negative control using antibodies alone without EVs showing large field of view and graph demonstrating the proportion of CD81 and CD63 antibodies.

**Figure 4 cells-10-02948-f004:**
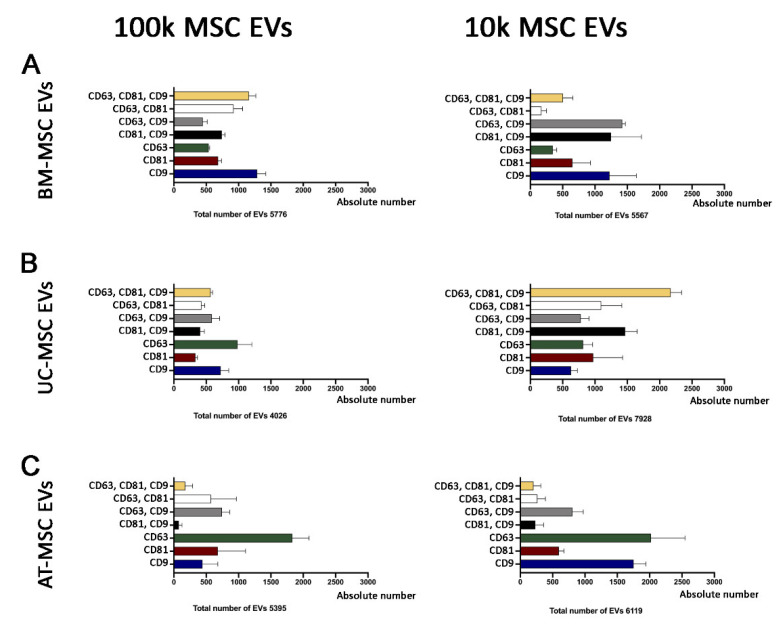
Super-resolution microscopy analysis of 100k and 10k MSC EVs. The graphs show triple positive, double positive and single positive EVs of 100k and 10k MSC EVs. (**A**) BM-MSC EVs, (**B**) UC-MSC EVs, (**C**) AT-MSC EVs. The total number of single EVs analysed is reported below each graph.

**Figure 5 cells-10-02948-f005:**
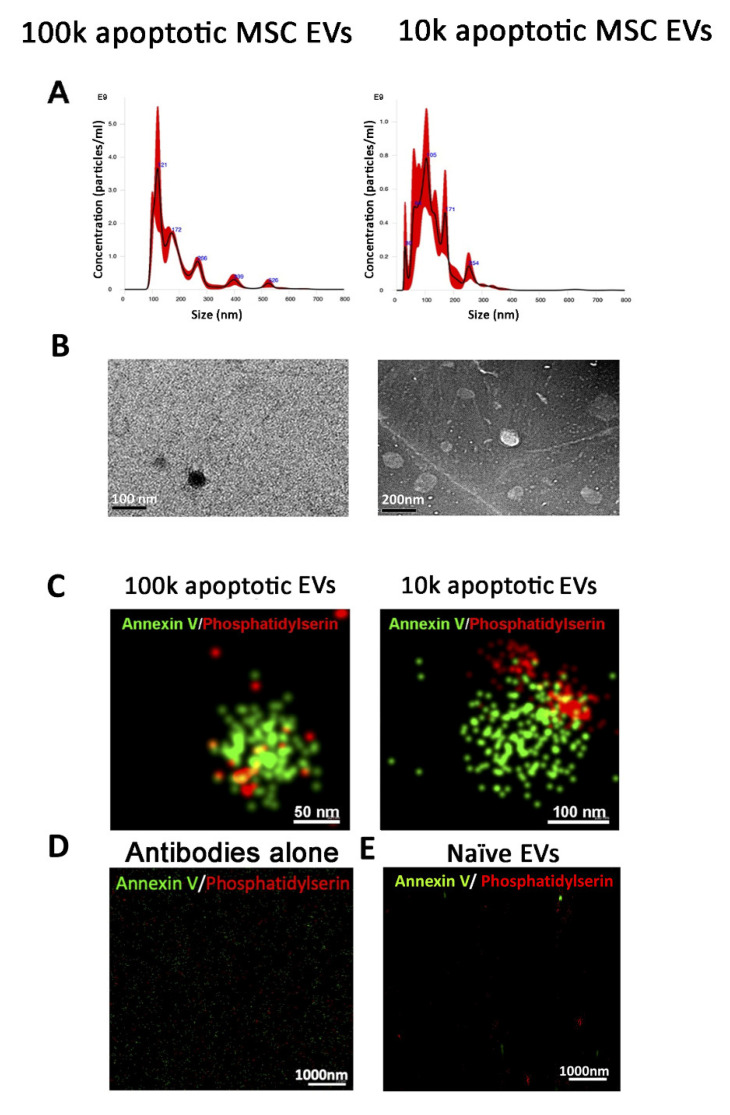
Characterisation of 100k and 10k apoptotic MSC EVs. (**A**) Representative graphs of nanoparticle tracking analysis of 100k apoptotic MSC EVs (left panel) and 10k apoptotic MSC EVs (right panel). (**B**) Representative images of transmission electron microscopy of 100k apoptotic MSC EVs (left panel) and 10k apoptotic MSC EVs (right panel). (**C**) Representative super-resolution microscopy images of 100k apoptotic MSC EVs (left panel) and 10k apoptotic MSC EVs (right panel) stained with Annexin V (green) and Phosphatidylserine (red). (**D**) Representative super-resolution microscopy image of Annexin V (green) and Phosphatidylserine (red) as a negative control without EVs. (**E**) Representative super-resolution microscopy image of 100k MSC EVs isolated from naive MSCs and stained with Annexin V (green) and Phosphatidylserine (red).

**Figure 6 cells-10-02948-f006:**
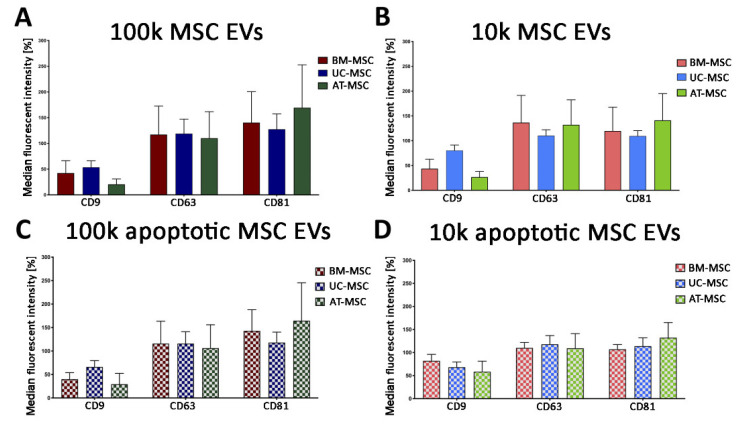
MACSPlex tetraspanin analysis of 100k MSC EVs and 10k MSC EVs of naive and apoptotic MSCs. Histograms represent the median fluorescence intensity of CD9, CD63 and CD81 tetraspanins for 100k MSC EVs (**A**,**C**) and 10k MSC EVs (**B**,**D**) isolated from naive (**A**,**B**) and apoptotic (**C**,**D**) MSCs. BM-MSC EVs, UC-MSC EVs and AT-MSC EVs were compared. Data are expressed as mean ± SD of three different experiments.

**Figure 7 cells-10-02948-f007:**
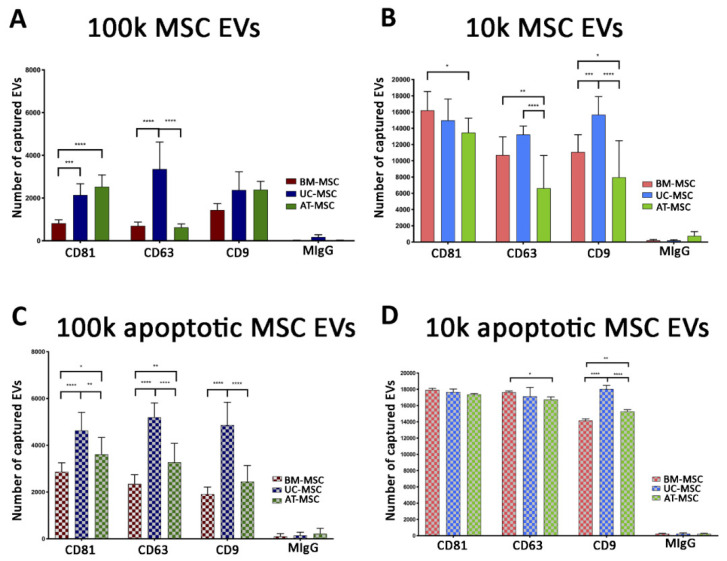
ExoView tetraspanin analysis of 100k and 10k MSC EVs of naive and apoptotic MSCs. Histograms represent the number of captured EVs for CD9, CD63 and CD81 tetraspanins and negative control. 100k MSC EVs (**A**,**C**) and 10k MSC EVs (**B**,**D**) isolated from naive (**A**,**B**) and apoptotic (**C**,**D**) MSCs were analysed. BM-MSC EVs, UC-MSC EVs and AT-MSC EVs were compared. Data are expressed as mean ± SD of three different experiments. A *p* value < 0.05 was considered significant (* *p* < 0.05, ** *p* < 0.001, *** *p* < 0.001, **** *p* < 0.0001).

**Figure 8 cells-10-02948-f008:**
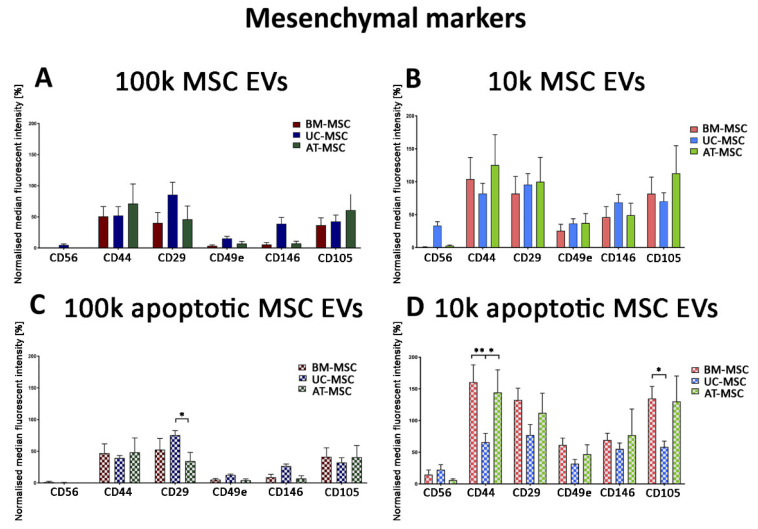
MACSPlex mesenchymal marker analysis of 100k and 10k MSC EVs of naive and apoptotic MSCs. (**A**–**D**) Histograms represent normalised fluorescence intensity of mesenchymal markers (CD56, CD44, CD29, CD49e, CD146, CD105) for 100k MSC EVs (**A**,**C**) and 10k MSC EVs (**B**,**D**) isolated from naive (**A**,**B**) and apoptotic (**C**,**D**) MSCs. BM-MSC EVs, UC-MSC EVs and AT-MSC EVs were compared. Data are expressed as median fluorescence intensity normalised to the median fluorescence intensity of tetraspanins ± SD of three different experiments. A *p* value < 0.05 was considered significant (* *p* < 0.05, ** *p* < 0.001).

**Figure 9 cells-10-02948-f009:**
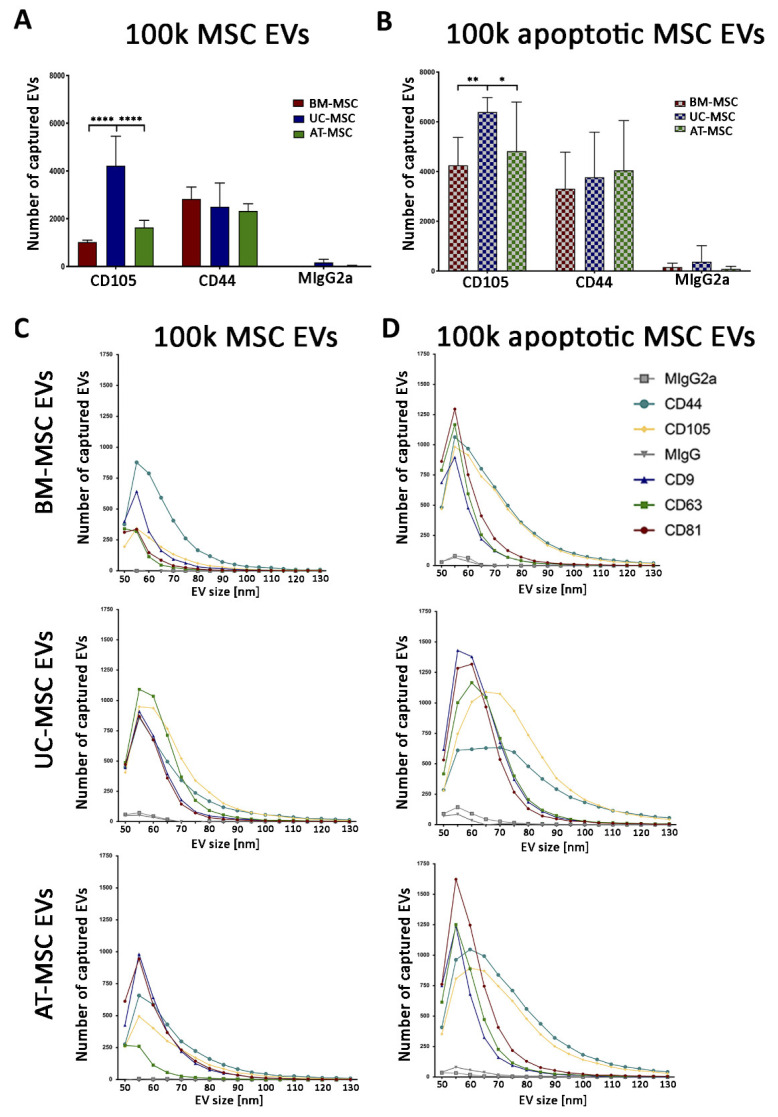
ExoView mesenchymal marker analysis of 100k of naive and apoptotic MSC EVs. (**A**,**B**) Histograms showing the number of captured EVs for mesenchymal markers (CD44, CD105) and negative control for 100k MSC EVs (**A**) and 100k apoptotic MSC EVs (**B**). Data are mean ± SD of three different experiments. A *p* value <0.05 was considered significant (* *p* < 0.05, ** *p* < 0.001, **** *p* < 0.0001). (**C**,**D**) Size of naive (**C**) and apoptotic (**D**) 100k MSC EVs from BM-MSC EVs, UC-MSC EVs and AT-MSC EVs.

**Figure 10 cells-10-02948-f010:**
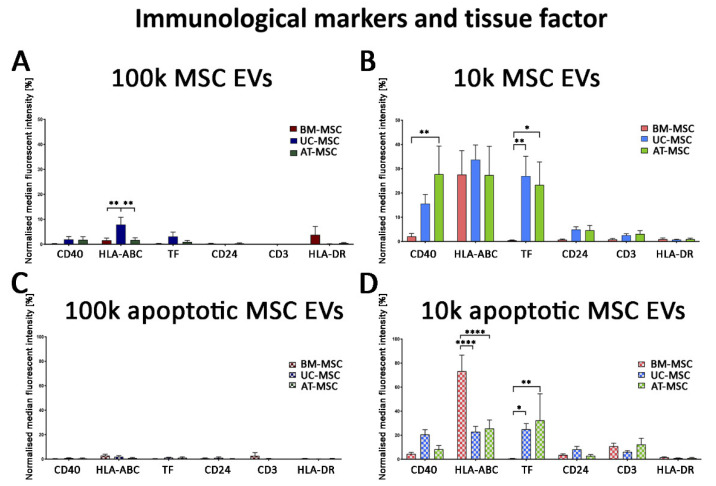
MACSPlex immunological marker analysis of 100k and 10k MSC EVs of naive and apoptotic MSCs. (**A**–**D**) Histograms represent normalised fluorescence intensity of immunity markers (CD40, HLA-ABC, CD24, CD3 and HLA-DR) and tissue factor (TF) for 100k MSC EVs (**A**,**C**) and 10k MSC EVs (**B**,**D**) isolated from naive (**A**,**B**) and apoptotic (**C**,**D**) MSCs. BM-MSC EVs, UC-MSC EVs and AT-MSC EVs were compared. Data are expressed as median fluorescence intensity normalised to the median fluorescence intensity of tetraspanins ± SD of three different experiments. A *p* value < 0.05 was considered significant (* *p* < 0.05, ** *p* < 0.001, **** *p* < 0.0001).

**Table 1 cells-10-02948-t001:** Average size and EVs concentration in 1 mL measured by Nanosight.

100k					10k			
Naive	Mode (nm)	SD	Concentration (Particles/mL)	SD	Mode (nm)	SD	Concentration (Particles/mL)	SD
BM	187.70	14.31	5.1 × 10^8^	2.4 × 10^8^	188.83	2.73	5.8 × 10^8^	3.1 × 10^8^
UC	184.23	19.63	3.6 × 10^8^	2.1 × 10^8^	197.07	8.54	6.8 × 10^8^	4.9 × 10^8^
AT	208.53	26.44	6.2 × 10^8^	2.5 × 10^8^	238.61	25.38	6.2 × 10^8^	2.7 × 10^8^
**100k**					**10k**			
**Apoptotic**	**Mode (nm)**	**SD**	**Concentration (Particles/mL)**	**SD**	**Mode (nm)**	**SD**	**Concentration (Particles/mL)**	**SD**
BM	183.67	15.63	1.4 × 10^9^	1.1 × 10^9^	169.10	18.71	1.4 × 10^9^	1.1 × 10^9^
UC	179.57	16.24	7.8 × 10^9^	9.0 × 10^9^	211.03	14.61	1.2 × 10^9^	1.4 × 10^9^
AT	170.90	35.94	1.1 × 10^9^	6.7 × 10^8^	202.77	22.80	7.9 × 10^9^	1.0 × 10^9^

**Table 2 cells-10-02948-t002:** Mean size of EVs measured by transmission electron microscopy, super-resolution microscopy and ExoView.

	Sample	TEM	Super-Resolution Microscopy	ExoView (50–200 nm Detection)
SIZE 100k EVs [nm]	BM	40–100	88.00 ± 7.94	61.85 ± 3.64
	UC	40–100	88.00 ± 3.46	59.14 ± 2.03
	AT	40–100	98.00 ± 2.11	59.92 ± 2.72
SIZE 10k EVs [nm]	BM	100–300	140.00 ± 5.77	90.00 ± 1.73
	UC	100–300	120.00 ± 5.77	90.33 ± 2.34
	AT	100–300	120.00 ± 8.42	92.83 ± 5.08

## Data Availability

Data supporting reported results can be obtained by the corresponding author under request.

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
