# Peer review of "Surface Marker Expression in Small and Medium/Large Mesenchymal Stromal Cell-Derived Extracellular Vesicles in Naive or Apoptotic Condition Using Orthogonal Techniques"

_cells, 2021, doi:10.3390/cells10112948_

Round 1
Reviewer 1 Report
several MSCs were used in the present study and this should be clearly stated. Why use different cells, any difference of EVs from those cells?
Author Response
To reviewer 1.
We thank the reviewer for the comment.
Several MSCs were used in the present study and this should be clearly stated. Why use different cells, any difference of EVs from those cells?
Answer. We are sorry that the aim of our study was unclear. We now better clarify the aim of the study and the rationality under the use of different MSC sources and their related EVs. MSCs can be isolated from various sources, all described to possess pro regenerative capacities. However, it is reported variability among MSC sources and every single donor in their biological effects. Therefore, this paper aimed to fully characterise and compare the profile of EVs from three different MSC sources (the most used), using three donors each, applying the same experimental conditions for MSC culture, EV isolation and analysis. For this comparison, we used different techniques following the standard and requirements of the ISEV community, including innovative single-EV analysis techniques such as ExoView and super-resolution microscopy. Standard culture and apoptotic conditions were applied in consideration to the possible effects of apoptotic EVs reported in the literature (ref. 18). Our results show that the characterisation profile of MSC EV fractions is consistent among different MSC sources. See pages 2 and 3, lines 84-90.
Reviewer 2 Report
The issue of the submitted work is for interest of a high audience and deals with a hot topic of the EVs field: the characterization of EVs from different metabolic stage cells. Nevertheless, before its publication, the following comments should be amended.
General comments:
- I see a lack of consistency between how you describe in the introduction that EVs must be characterize and what you characterize in the results. In the introduction (lines 66-67), you state that composition and function of the EVs must be described to differentiate different subpopulations, but you did not do any functional test. In fact, according to MISEV2018, it is a must to add these functionality test. You should add them before publication.
- In relationship to the previous comment, you MUST add information about MISEV guidelines in the introduction! This is mandatory, to demonstrate that you work with the recognized guidelines from the experts on the field.
- Using primary cell lines requires the approval of ethical committees. Please, add this information.
- In general, TEM micrographies are very different between 100k and 10k samples; I am referring to the background of the image. Please, make a comment on that.
- In the discussion, references to MISEV need to be added too. In general, you have followed their guidelines, but clarify that on the text.
- The use of biorender for figures needs to be stated.
Specific comments:
- Lines 45-47 EVs are also classified depending on their biogenesis. Please, add this.
- In NTA description, you state you performed the measures diluting samples in water, but this cannot be done, since you would be producing an osmotic imbalance that would result in destructuration of the EVs. Please, comment on that.
- In cytofluorimetric analysis description, the panel with the fluorescent dyes / antibodies used, must be added.
- Figure 2. NTA data are not correct. The y axis lacks the 10 to the XXX; I mean that you should be measuring much more orders of magnitude to fit with the Table data. Please, amend this error.
- In Table 2, TEM of 100k vesicles must be given. <100 is not enough, since TEM has enough resolution below 100 nm to enable the size measurement.
- Figure 6: try to use plots of the same size in a same figure.
Author Response
To reviewer 2.
We thank the reviewer for the comments and we did our best to reply accordingly.
I see a lack of consistency between how you describe in the introduction that EVs must be characterize and what you characterize in the results. In the introduction (lines 66-67), you state that composition and function of the EVs must be described to differentiate different subpopulations, but you did not do any functional test. In fact, according to MISEV2018, it is a must to add these functionality test. You should add them before publication.
Answer. Thank you for your comment. We agree with the reviewer that the functionality of EVs is of importance. However, the present study focused on one aspect of EV characterization, which is the analysis. As the ISEV community strongly recommends that a robust EV characterization be performed, our study aims to address this point and provide results on the consistency of the different techniques widely used for EV characterization. For this aim, we compared the profile of EVs (both 10k and 100k EVs) from three different MSC sources, using three donors each, applying the same experimental conditions for MSC culture, EV isolation and analysis. For this comparison, we used different techniques following the standard and requirements of the ISEV community, including innovative single-EV analysis techniques such as ExoView chip and super-resolution microscopy. Standard culture and apoptotic conditions were applied. This also generated a large number of samples (6 for naive and 6 for apoptotic conditions, with three donors for each source) that were compared with a list of techniques. The comparison of the EV functionality will be the second aim of our consortium, and considering the number of different samples to compare, this will require extensive effort and time. We added a phrase in the discussion reporting the need for functional analysis to describe EV characteristics fully. Page 17, lanes 412-413.
In relationship to the previous comment, you MUST add information about MISEV guidelines in the introduction! This is mandatory, to demonstrate that you work with the recognized guidelines from the experts on the field.
Answer. Thank you for your comment. We now better clarify the MISEV recommendations, coupled with the recommendations for the identification of small therapeutic MSC-EVs (refs 7, 16, 17), lines 68-71.
Using primary cell lines requires the approval of ethical committees. Please, add this information.
Answer. Thank you for your comment. We now better clarify the point. Page 3, lines 94-101.
In general, TEM micrographies are very different between 100k and 10k samples; I am referring to the background of the image. Please, make a comment on that.
Answer. Thank you for your comment. We used and inserted in the Figure raw images, performed using different magnifications and in various TEM acquisitions, without post-production changes, which may affect the differences in background grayscale.
In the discussion, references to MISEV need to be added too. In general, you have followed their guidelines, but clarify that on the text.
Answer. Thank you for your comment. We now better clarify the point and add the MISEV ref. in lines 415.
The use of biorender for figures needs to be stated.
Answer. Thank you for your comment. We now specify the use of BioRender for Figure 1 in the legend.
Lines 45-47 à EVs are also classified depending on their biogenesis. Please, add this.
Answer. Thank you for your suggestion. We now add this EV classification, see lines 48-49.
In NTA description, you state you performed the measures diluting samples in water, but this cannot be done, since you would be producing an osmotic imbalance that would result in destructuration of the EVs. Please, comment on that.
Answer. Thank you for pointing out this mistake. It was an error as the samples were diluted in physiologic solution. We changed the text with the correct protocol. See lines 129-130.
In cytofluorimetric analysis description, the panel with the fluorescent dyes/antibodies used, must be added.
Answer. Thank you for your comment. We now better explain in the methods all the Abs and dyes provided within the MacsPlex kit, and we better detail the methodological procedures of the stainings. See page 4, lines 143-150.
Figure 2. NTA data are not correct. The y axis lacks the 10 to the XXX; I mean that you should be measuring much more orders of magnitude to fit with the Table data. Please, amend this error.
Answer. Thank you for noticing this mistake; we cut by accident the elevation indication (E8). We now replaced the full y-axis in the related Figures. See Figures 2 and 4.
In Table 2, TEM of 100k vesicles must be given. <100 is not enough, since TEM has enough resolution below 100 nm to enable the size measurement.
Answer. Thank you for your comment. We now better specify the size range of 100k EVs obtained by TEM in Table 2 (40-100 nm).
Figure 6: try to use plots of the same size in a same figure.
Answer. Thank you for your comment regarding the graphical aspect of this Figure. We aligned the graphs.
Round 2
Reviewer 2 Report
I see you have ammended all my comments, so I find the manuscript adequate to be published in the current form.